# Isolation and Genomic Characteristics of Cat-Borne *Campylobacter felis* sp. nov. and Sheep-Borne *Campylobacter ovis* sp. nov

**DOI:** 10.3390/microorganisms11040971

**Published:** 2023-04-08

**Authors:** Hairui Wang, Ying Li, Yixin Gu, Guilan Zhou, Xiaoli Chen, Xin Zhang, Zhujun Shao, Jianzhong Zhang, Maojun Zhang

**Affiliations:** 1State Key Laboratory for Infectious Disease Prevention and Control, National Institute for Communicable Disease Control and Prevention, Chinese Center for Disease Control and Prevention, Rd155, Changbailu, Changping, Beijing 102206, China; 2Shunyi District Center for Disease Control and Prevention, Beijing 101320, China

**Keywords:** *Campylobacter felis*, *Campylobacter ovis*, novel species, genomic characteristics, phylogenetic analyses

## Abstract

Nine novel bacterial strains were isolated from the feces of cats and sheep in 2019 and 2020 in Beijing, China. Cells were 1–3 μm long and ≤0.5 μm wide, Gram-stain negative, microaerobic, motile, oxidase positive, and urease negative. Phylogenetic analyses based on 16S rRNA gene sequences indicated that these nine isolates belong to the genus *Campylobacter* but formed two robust clades that were clearly separate from the currently recognized species and, respectively, isolated from the cat and sheep. Both these strains shared low 16S rRNA gene sequence similarity, dDDH relatedness, and ANI values with their closest species *C. upsaliensis* CCUG 14913^T^ and *C. lanienae* NCTC 13004^T^, and against each other, which are below the cut-off values generally recognized for isolates of the same species. The genomic DNA G + C contents of type strains XJK22-1^T^ and SYS25-1^T^ were 34.99 mol% and 32.43 mol%, respectively. Electron microscopy showed that these cells were spiral shaped, with bipolar single flagella. Based on results from genotypic, phenotypic, phylogenetic, and phylogenomic analyses, these nine strains represent two novel species within the genus *Campylobacter*, for which the names *Campylobacter felis* sp. nov. (Type strain XJK22-1^T^ = GDMCC 1.3684^T^ = JCM 35847^T^) and *Campylobacter ovis* sp. nov. (Type strain SYS25-1^T^ = GDMCC 1.3685^T^) are proposed.

## 1. Introduction

The genus *Campylobacter* belongs to the family Campylobacteraceae and the order Campylobacterales, which currently contains 37 and 13 validly described species and subspecies, respectively, and 2 not validly described species (https://lpsn.dsmz.de/genus/campylobacter) (accessed on 15 February 2023). Members of the *Campylobacter* genus are morphologically diverse, and can be spiral, curved, or rod shaped. These bacteria are nutritionally fastidious and grow under strictly anaerobic or microaerobic conditions. They naturally colonize humans, other mammals, birds, reptiles, shellfish, etc. [1,2,3].

*Campylobacter* is among the four main causes of gastroenteritis worldwide [4]. Most reported *Campylobacter* infections are caused by *C. jejuni*, which is a leading cause of bacterial gastroenteritis in humans worldwide [2] and whose antecedent infection could trigger a Guillain–Barré Syndrome (GBS) outbreak [5], and, to a lesser extent, *C. coli*, which accounts for 1–25% of all *Campylobacter*-related diarrheal diseases [2]. However, the other emerging *Campylobacter* pathogens are gaining increasing recognition as important pathogens in humans and animals [6]. In addition, with the continuous progress of isolation and culture technology, more and more novel *Campylobacter* species have been identified in recent years [7,8,9,10,11,12,13,14]. Domestic animals, especially pets, have close contact with humans. Additionally, *Campylobacter* spp. infections in humans caused by pets have been reported several times in recent years [15,16,17,18,19]. A total of 117 of 121 patients with *Campylobacter* infections reported contact with a dog in the week before symptom onset in 2016–2020 [20].

In this study, taxonomic and genomic characteristics of the novel *Campylobacter*-like isolates were described, and the phylogenetic relationships between the isolated strains and their closest relatives were also clarified. Based on polyphasic taxonomic analyses, these novel isolates are proposed as two novel *Campylobacter* species, designated *Campylobacter felis* sp. nov. (XJK22-1^T^, XJK33-1, XJK49-2, XJK56-3, XJK62-3, and XJK7-1) and *Campylobacter ovis* sp. nov. (SYS25-1^T^, SYS28-3, and S13-1).

## 2. Materials and Methods

### 2.1. Sampling, Isolation and Culturing

During the investigation of the *Campylobacter* spp. diversity in both healthy animals and animals with diarrhea, which included animals such as dogs, cats, sheep, and pigs that are in close contact with humans, isolation was carried out using the *Campylobacter* isolation kit incorporating a membrane filter method (ZC-CAMPY-002, Qingdao Sinova Biotechnology Co., Ltd., Qingdao, China). Briefly, 0.4 mL stool specimen suspension was transferred into a 4 mL enrichment buffer, which was provided in the kit. The principal component of the enrichment buffer was the modified Preston broth, which was described in the manual book. The enriched suspension was incubated at 37 °C for 24 h in a microaerophilic atmosphere consisting of 5% O_2_, 10% CO_2_, and 85% N_2_. About 300 μL cultured enrichment suspension was then spotted on the surface of the filter pasted on the double medium plates, which contained Karmali and Columbia agar, respectively, with 5% defibrinated sheep blood. The medium plates were incubated in a microaerophilic atmosphere at 37 °C for 48 h [21]. The suspected monoclonal colonies were selected and purified and were subjected to preliminary characterization by PCR amplification and sequencing of 16S rRNA gene sequence analysis [22], and subsequently conserved at −80 °C in BHI with 20% (*v/v*) glycerol for further identification. The exact 16S rRNA gene PCR amplification primer sequences were listed in Table 1.

### 2.2. Morphological, Physiological and Biochemical Characteristics

For the study of the morphological and biochemical characteristics, cells were cultivated and harvested in the late-exponential growth phase. Gram staining was conducted using a Gram-staining kit (Baso, Zhuhai, China) [23] and observed under a light microscope. The morphological characteristics of these two putative novel species’ type strains were determined using transmission electron microscopy. Fresh cells were gently suspended in 0.1 M phosphate-buffered saline (PBS) to an OD600 of 1 and collected via gentle centrifugation. The pellet was gently resuspended in a 2% (*v*/*v*) glutaraldehyde solution for fixation. Fixation was completed by incubating strains for 1 h on the grid. All samples were stained with 2% (*w*/*v*) uranyl acetate for 1 min and examined using a Hitachi H7700 transmission electron microscope (Eclipse Ci-L, NIKON, Tokyo, Japan) at 80 kV.

The catalase activity was evaluated using a 3% (*v*/*v*) H_2_O_2_ solution for bubble production. The general phenotypic traits of *Campylobacter* spp., oxidase, catalase, hydrolysis of hippurate, and indoxyl acetate were evaluated using the *Campylobacter* Biochemical Assay kit (ZC-CAMPY-010, Qingdao Sinova Biotechnology Co., Ltd., Qingdao, China). Further biochemical characteristics were obtained using the identification system of API Campy, strictly following the manufacturers’ instructions (bio-Mérieux, Lyon, France). Biochemical tests were carried out to characterize the physiology and chemotaxonomy of the isolates. *C. jejuni* ATCC 33560^T^, *C. coli* ATCC 33559^T^, and *C. upsaliensis* CCUG 14913^T^ were used as controls.

### 2.3. Antimicrobial Susceptibility Testing

The minimum inhibitory concentrations (MICs) for 11 antimicrobials (erythromycin, azithromycin, nalidixic acid, ciprofloxacin, gentamicin, streptomycin, chloramphenicol, florfenicol, tetracycline, telithromycin, and clindamycin) at concentrations ranging from 0.02 to 256 μg mL^−1^ were determined for all isolates using the agar dilution method (ZC-AST-001, Qingdao Sinova Biotechnology Co., Ltd., Qingdao, China) and the gradient strip diffusion method (E-test, bio Mérieux, Nürtingen, Germany) following the manufacturer’s instructions, as previously reported [24,25]. The MIC was read as the lowest concentration without visible growth. Type strain *C. jejuni* ATCC 33560^T^ was used as a control.

### 2.4. Species-Specific PCR

To define a diagnostic method for the rapid detection and identification of these two putative new species, specific TaqMan real-time PCR primers targeting different genes were designed. The exact primer and probe sequences are listed in Table 1. The PCR conditions were as follows: initial denaturation at 94 °C for 30 s, followed by 40 cycles of 94 °C for 5 s and 60 °C for 30 s. The positive and negative results refer to the cycle threshold (Ct) value of the real-time PCR that, according to the manual, Ct < 35 with the typical S curve, is determined as positive; Ct > 40 or no typical S curve is determined as negative, and 35 ≤ Ct ≥ 40 should be repeated 3 times and determined as weakly positive after the third repeat. To assess the sensitivity and specificity of this qPCR method, some species of *Campylobacter* strains which were stored in our laboratory, including *C. coli*, *C. concisus*, *C. fetus* subsp. *fetus*, *C. gracilis*, *C. helveticus*, *C. hyointestinalis* subsp. *hyointestinalis*, *C. jejuni* subsp. *doylei*, *C. jejuni* subsp. *jejuni*, *C. lari* subsp. *lari*, *C. mucosalis*, *C. rectus*, *C. showae*, and *C. upsaliensis*, were used as reference strains. Several kinds of other genus bacteria stored in our laboratory were used as negative controls, such as *Arcobacter butzleri*, *Arcobacter skirrowii*, *Helicobacter pylori*, *Escherichia coli*, etc.

### 2.5. Genome Extraction and Sequencing

After culturing, the DNA for genome sequence was extracted using the QIAamp DNA Mini Kit (Qiagen, German) according to the manufacturer’s instructions for sequencing. Then, the NanoDrop 2000 spectrophotometer (Thermo Fisher, Massachusetts, USA) was used to measure the concentration and purity of DNAs. The quality requirements were a concentration ≥ 20 ng/μL and a total amount > 2 μg. The purity requirement was as follows: OD260/OD280 value should be between 1.6 and 1.8. The DNA sequencing was performed by an Illumina PE150 platform (Illumina Inc., San Diego, CA, USA) at the Novogene Corporation (Beijing, China) with a depth of 100× coverage. To sequence the genomes, a 350 bp paired-end library was constructed and then 150 bp reads were generated. FastQC v0.11.8 and fastp v0.23.2 were applied to evaluate and improve the quality of the raw sequence data, respectively. Low-quality reads were removed if the quality scores of ≥3 consecutive bases were  ≤Q30. The clean reads were assembled by SOAPdenovo v2.40.

### 2.6. Genomic Analysis

The genome was predicted and annotated using the Prokka pipeline [26] and tRNA-scan tool [27]. Phage Search Tool (PHAST) web server (http://phaster.ca/) (accessed on 29 January 2023)and phiSpy v4.2.21 [28] were used to search for phage sequences. The antimicrobial resistance genes were predicted using the Comprehensive Antibiotic Resistance Database (CARD) [29] and the ResFinder v4.2.21. The virulence genes of all the genomes were detected on VFanalyzer (http://www.mgc.ac.cn/cgi-bin/VFs/v5/main.cgi?func=VFanalyzer) (accessed on 7 September 2022) [30]. The digital DNA-DNA hybridization (dDDH) relatedness was calculated and compared using the Genome-to-Genome Distance Calculator 3.0 (https://ggdc.dsmz.de/) (accessed on 15 February 2023) [31]. The average nucleotide identity (ANI) values were determined by pyani 0.2.10 [32].

### 2.7. Phylogenetic and Phylogenomic Analysis

To determine the phylogenetic positions of strains, 16S rRNA gene PCR amplification was performed with primers 27F and 1492R, as previously reported. Each almost-complete sequence of the 16S rRNA gene PCR product was purified, sub-cloned into the pMD18-T vector for 30 min at 16 °C, transformed into *Escherichia coli* DH5α, and the inserted 16S rRNA gene fragment was obtained from a single colony after lysis and sequenced. The newly generated 16S rRNA gene sequences were compared with other *Campylobacter* species by EzBioCloud’s identification service to locate their taxonomic position [33]. Multiple sequence alignment of the 16S rRNA gene sequences of the type strains in the genus *Campylobacter* was performed using the MAFFT 7.471 software [34] and phylogenetic analysis using the software package MEGA X [35], by the neighbor-joining (NJ) [36], maximum parsimony (MP) [37] and maximum likelihood (ML) [38] algorithms with a bootstrap analysis of 1000 replicates [39] and strain *Arcobacter butzleri* ATCC 49616^T^ was used as an outgroup.

The protein sequences of core genes of genomes from the isolates and the other *Campylobacter* species were extracted using the CD-HIT v4.8.1 [40] based on 40% protein sequence similarity and aligned to reconstruct a phylogenomic tree using FastTree v2.1.11 [41]. The multiple sequence alignment of the core genomes of the genus *Campylobacter* was also performed using the MAFFT software. Then, the phylogenomic tree was visualized by Dendroscope 3.8.3 [42], and modified with Interactive Tree of Life (https://itol.embl.de/) (accessed on 28 February 2023).

### 2.8. Accession Numbers

The GenBank/EMBL/DDBJ accession numbers for the nearly full-length 16S rRNA gene and the draft genome sequences of these 9 isolates were submitted to NCBI (https://www.ncbi.nlm.nih.gov/) (accessed on 9 August 2022). In addition to these 9 isolates, the other 10 temporarily undefined genomes in NCBI which belonged to *Campylobacter ovis* sp. nov. and genomes from the other type strains of *Campylobacter* genus were downloaded from NCBI. There were no genomes found in NCBI belonging to *Campylobacter felis* sp. nov. More information about the genomes of novel species used in this study was listed in Table 2.

## 3. Results and Discussion

### 3.1. Isolation and Phenotypic Characterization

Six isolates (XJK22-1^T^, XJK33-1, XJK49-2, XJK56-3, XJK62-3, and XJK7-1) and three other isolates (S13-1, SYS25-1^T^, and SYS28-3) were isolated from fecal samples of asymptomatic carriers in cat and sheep, respectively.

These Gram-negative, microaerobic, motile, spiral-shaped cells with bipolar single flagellum ranged from 1.8 to 2.2 μm for type strain XJK22-1^T^ and 2.1 to 2.5 μm for type strain SYS25-1^T^ (Figure 1). Colonies were circular, 2–3 mm in diameter, smooth, and gray after 2 days of growth on Karmali agar with 5% defibrinated sheep blood. The cells appeared coccoid after 5–6 days of incubation or when exposed to air.

Like most other *Campylobacter* species, the isolates were positive for oxidase and negative for urease activities. The catalase was negative for species of type strain XJK22-1^T^ and positive for species of type strain SYS25-1^T^. While most *Campylobacter* species are unable to hydrolyze hippurate and able to hydrolyze indoxyl acetate and reduce nitrate, two-thirds (*n* = 6, XJK22-1^T^, XJK33-1, XJK49-2, XJK56-3, XJK62-3, and XJK7-1) of the new species isolates were able to hydrolyze hippurate and indoxyl acetate positive and could reduce nitrate, and the other one-third (*n* = 3, S13-1, SYS25-1^T^, and SYS28-3) could not hydrolyze hippurate and indoxyl acetated, among which only two isolates could reduce nitrate (*n* = 2, S13-1 and SYS25-1^T^). There are only five other species of the genus *Campylobacter*, *C. avium*, *C. curvus*, *C. jejuni*, *C. hepaticus*, and *C. geochelonis* that can hydrolyze hippurate [11].

All nine isolates were initially identified as *C. upsaliensis* and *C. lanienae* according to the results of the sequencing of the 16S rRNA gene. Thus, it was not unexpected that the results of the standard biochemical of strain XJK22-1^T^ and SYS25-1^T^ showed a strong similarity between the composite phenotypic profile observed from the isolates and the phenotypic profile reported previously for *C. upsaliensis* CCUG 14913^T^ and *C. lanienae* NCTC 13004^T^, respectively (Table 3). Nevertheless, these six strains (XJK22-1^T^, XJK33-1, XJK49-2, XJK56-3, XJK62-3, and XJK7-1) could be unambiguously distinguished from *C. upsaliensis* by the ability to hydrolyze hippurate, which may challenge the discriminability of *C. jejuni* to distinguish from other *Campylobacter* species by hydrolyzing hippurate. Meanwhile, the other three strains (S13-1, SYS25-1^T^, and SYS28-3) could be distinguished from *C. lanienae* by the phenotypic characteristic of positive for GGT (gamma-glutamyltransferase) and negative for reduction of TTC (Triphenyltetrazolium chloride). These partial results preliminarily supported the theory that these nine strains were two novel *Campylobacter* species.

### 3.2. Phylogenetic and Phylogenomic Analysis

The comparison against the EzTaxon-e database of near full-length 16S rRNA gene sequences (1474–1478 bp) revealed that our 9 isolates and the other 10 strains were most closely related to the representatives of the genus *Campylobacter* (Domain, Bacteria; Phylum, Pseudomonadota; Class, Epsilonproteobacteria; Order, Campylobacterales; Family, Campylobacteraceae). Strains XJK22-1^T^, XJK33-1, XJK49-2, XJK56-3, XJK62-3, and XJK7-1 were closest to *C. upsaliensis* CCUG 14913^T^ (96.66% of 16S rRNA gene identity of strain XJK22-1^T^), whereas strains S13-1, SYS25-1^T^, SYS28-3, and the other 10 strains were closest to *C. lanienae* NCTC 13004^T^ (98.47% of 16S rRNA gene identity of strain SYS25-1^T^). The similarity between strains XJK22-1^T^ and SYS25-1^T^ was 90.70%. These values were lower than 98.70%, which was the generally accepted threshold for species [43], suggesting that these 19 strains should belong to the genus *Campylobacter* and represent two novel species.

The NJ phylogenetic tree (Figure 2) based on the nearly complete 16S rRNA gene sequences also revealed that these 19 strains belong to the genus *Campylobacter* and form two independent clusters. Strains, XJK22-1^T^, XJK33-1, XJK49-2, XJK56-3, XJK62-3, and XJK7-1, were grouped into a cluster and closest to *C. upsaliensis* CCUG 14913^T^, *C. vulpis* 251/13^T^, *C. helveticus* ATCC 51209^T^, whereas strains S13-1, SYS25-1^T^, SYS28-3, and the other 10 strains were grouped into another cluster and closest to *C. lanienae* NCTC 13004^T^, *C. infantis* 19S00001^T^, *C. hyointestinalis* subsp. *lawsonii* CCUG 34538^T^, which was similar to the topological results obtained from the ML and MP trees (Appendix A).

Based on 40% protein identity, orthologous groups of 332 core genes shared by our 9 isolates and all available genomes of the genus *Campylobacter* were extracted and used to build a phylogenomic tree (Figure 3).

This robust pangenomic tree revealed that strains XJK22-1^T^, XJK33-1, XJK49-2, XJK56-3, XJK62-3, and XJK7-1 were grouped with *C. upsaliensis* CCUG 14913^T^ and strains S13-1, SYS25-1^T^, and SYS28-3 were grouped with *C. lanienae* NCTC 13004^T^, a result identical to that of phylogenetic trees based on 16S rRNA gene sequences, further proving that the isolates belong to the genus *Campylobacter*.

### 3.3. Genome Characteristics

The draft genome of strain XJK22-1^T^ (1.70 Mb) was predicted to contain 1747 coding genes and carried 2 rRNA genes and 43 tRNA genes, whereas SYS25-1^T^ (1.58 Mb) was predicted to contain 1567 coding genes and carried 1 CRISPR/Cas loci, which were obviously different between two type strains, 1 rRNA gene, and 38 tRNA genes. The genomic DNA G + C content of type strain XJK22-1^T^ was 34.99 mol%, which is slightly higher than the most closely related bacterium, *C. upsaliensis* CCUG 14913^T^ (34.73%). Meanwhile, strain SYS25-1^T^ was 32.43 mol%, which is lower than the most closely related bacterium, *C. lanienae* NCTC 13004^T^ (34.60%), and within the range of DNA base compositions previously reported for the members in the genus *Campylobacter* (29–47 mol% G + C) [44]. This further confirms that these nine strains are two novel *Campylobacter* species. More genomic characteristics (G + C content, CDS, size, etc.) are listed in Table 4.

The dDDH scores within each strain pair were 73.80–94.00% (XJK22-1^T^, XJK33-1, XJK49-2, XJK56-3, XJK62-3, and XJK7-1) and 75.00–99.60% (strains S13-1, SYS25-1^T^, SYS28-3, and the other 10 strains), which were well above 70%, the threshold for species demarcation. In contrast, the scores of these 19 strains with their closest species were below 70%. Meanwhile, the ANI values within each strain pair were 96.92–99.26% (strains XJK22-1^T^, XJK33-1, XJK49-2, XJK56-3, XJK62-3, and XJK7-1) and 97.09–99.91% (strains S13-1, SYS25-1^T^, SYS28-3, and the other 10 strains), in contrast to below 95%, the cutoff for species demarcation, between our isolates and all established species of *Campylobacter* (Table 5 and Appendix A). Based on the gold standard for the delineation of bacterial species [45], these results suggested that strains XJK22-1^T^ and SYS25-1^T^ represented two novel species of the genus *Campylobacter*.

### 3.4. Antibiotic Resistance and Pathogenicity

Antibiotic resistance demonstrated that the strains XJK22-1^T^, XJK33-1, XJK49-2, XJK56-3, XJK62-3, and XJK7-1 were resistant to three types of antibiotics, macrolides (erythromycin (MIC, ≥64 μg mL^−1^) and azithromycin (MIC, ≥64 μg mL^−1^)), quinolones (nalidix acid (MIC, ≥32 μg mL^−1^)), aminoglycosides (streptomycin (MIC, ≥64 μg mL^−1^)), and yet strains S13-1, SYS25-1^T^, and SYS28-3 were resistant to quinolones (nalidix acid (MIC, ≥16 μg mL^−1^) and ciprofloxacin (MIC, ≥8 μg mL^−1^)). Part of our results is consistent with a previous report showing that *Campylobacter* species are highly resistant to quinolones [25,46].

In genomes of XJK22-1^T^, XJK33-1, XJK49-2, XJK56-3, XJK62-3, and XJK7-1, the aminoglycoside antibiotic resistance gene *APH(2”)-If* was found in two out of six strains (33.33%, XJK49-2 and XJK62-3). Point mutations in the 23S ribosomal RNA domain, *Cjej_23S_ERY* (A2075G), which confer resistance to macrolide antibiotics, was identified in all six strains (100.00%), and the prevalence of *Cjej_gyrA_FLO* (T86I) conferring resistance to fluoroquinolones was found among these six strains (100.00%). Meanwhile, in genomes of S13-1, SYS25-1^T^, SYS28-3, and the other 10 strains, the prevalence of *Cjej_gyrA_FLO* (T86I) was also identified in 2 of 13 strains (15.38%, SYS25-1^T^ and SYS28-3), and the tetracycline and fluoroquinolone antibiotic resistance gene *adeF* was found in 11 of 13 strains (84.62%, S13-1, SYS25-1, SYS28-3, RM8835, RM8965, RM9262, S0112, RM9263, RM9261, RM12175, RM8964). These antibiotic resistance genes were consistent with their resistance phenotype, respectively.

In genomes of these strains, numerous *Campylobacter* virulence-associated genes were detected, which could encode genes related to adherence, colonization and immune evasion, invasion, motility and export apparatus, secretion system, and toxins. The species of type stains of XJK22-1^T^ and SYS25-1^T^ had significantly different virulence-associated gene profiles. The species of type stain XJK22-1^T^ have more *Campylobacter* virulence-associated genes than the species of SYS25-1^T^. The species of XJK22-1^T^ have the complete cytolethal distending toxin, while the species of type strain SYS25-1^T^ only have an incomplete type Ⅳ secretion system (T4SS). The details of the virulence genes were presented in Figure 4.

### 3.5. Specific Real-Time PCR

These nine strains were simultaneously tested using Taqman real-time PCR to identify the species of *Campylobacter*. Amplification of six strains, XJK22-1^T^, XJK33-1, XJK49-2, XJK56-3, XJK62-3, and XJK7-1, was positive for *Campylobacter felis* sp. nov. specific qPCR, whereas the other *Campylobacter* species and blank control were qPCR negative. Similarly, S13-1, SYS25-1^T,^ and SYS28-3 were positive for *Campylobacter ovis* sp. nov. specific qPCR, and the other *Campylobacter* species and blank control were qPCR negative. This suggests that these two specific real-time PCR are robust and can be used to rapidly discriminate between these two novel species and other *Campylobacter* species strains.

## 4. Conclusions

A polyphasic approach, including DNA sequencing and analysis (16S rRNA and whole-genome sequencing), electron microscopy, and a wide range of biochemical tests, as suggested by On et al. [47], provided sufficient evidence to distinguish these nine isolates from their closest related type strains and to confirm that they represent two novel species. With XJK22-1^T^ and SYS25-1^T^ as the type strains, we suggest the names *Campylobacter felis* sp. nov. and *Campylobacter ovis* sp. nov. for the two novel members of the genus *Campylobacter*.

### 4.1. Description of Campylobacter felis sp. nov.

#### *Campylobacter felis* (*fe’lis.* L. gen. n. felis of a Cat)

The cells are Gram negative, mesophilic, motile, and spiral shaped with sizes between 1.8–2.2 μm after 48 h of growth on Karmali or Columbia agar with 5% defibrinated sheep blood in a microaerophilic atmosphere at 37 °C. The colonies are wet, flat, gray, circular, and smooth, with sizes between 1.0 and 1.3 mm after 2 days of growth, but they may vary in size and morphology after a long incubation. No hemolysis on blood agar is observed. Fresh cells are motile and have long bipolar single flagella.

These six strains are negative for catalase, urease, GGT, TTC, and pyroglutamyl-peptidase I, and cannot produce H_2_S. On the other hand, they are positive for oxidase, reduced nitrate, L-arginine arylamidase, L-aspartic acid arylamidase, and alkaline phosphatase, and all can hydrolyze indoxyl acetate and hippurate. All six strains are sensitive to chloramphenicol, florfenicol, and tetracycline and resistant to erythromycin, azithromycin, nalidixic acid, and streptomycin.

The type strain XJK22-1^T^ (=GDMCC 1.3684^T^ = JCM 35847^T^), which was isolated from the feces of cats in 2019 and 2020 in Beijing, China, has a DNA G + C content of 34.99 mol%. The other strains XJK33-1, XJK49-2, XJK56-3, XJK62-3, and XJK7-1 are also classified in this species.

### 4.2. Description of Campylobacter ovis sp. nov.

#### *Campylobacter ovis* (*o’vis.* L. gen. n. ovis of a Sheep)

The cells are Gram negative, mesophilic, motile, and spiral or S-shaped with sizes between 2.1–2.5 μm after 48 h of growth on Karmali or Columbia agar with 5% defibrinated sheep blood in a microaerophilic atmosphere at 37 °C. The colonies are wet, flat, gray, circular, and smooth, with sizes between 1.0 and 1.3 mm after 2 days of growth, but they may vary in size and morphology after a long incubation period. No hemolysis on blood agar is observed. Fresh cells are motile and have long bipolar single flagella.

These three strains are negative for urease, TTC, pyroglutamyl-peptidase I, and L-aspartic acid arylamidase and cannot hydrolyze indoxyl acetate and hippurate or produce H_2_S. On the other hand, they are positive for catalase, oxidase, GGT, L-arginine arylamidase, and alkaline phosphatase and are variable for the reduction of nitrate. All three strains are resistant to nalidixic acid and sensitive to erythromycin, azithromycin, gentamicin, streptomycin, chloramphenicol, florfenicol, tetracycline, telithromycin, and clindamycin.

The type strain SYS25-1^T^ (=GDMCC 1.3685^T^), which was isolated from the feces of sheep in 2019 and 2020 in Beijing, has a DNA G + C content of 32.43 mol%. The other strains, S13-1 and SYS28-3, are also classified in this species.

## 5. Limitations

In this study, we utilized a polyphasic approach to isolate and identify two novel *Campylobacter species*, which we have named *Campylobacter felis* sp. nov. and *Campylobacter ovis* sp. nov. Although their pathogenicity is currently unknown, studies indicate that further investigation into the potential health implications of these novel bacteria is necessary. Such research may aid in the management of diseases that could potentially be caused by these novel *Campylobacter* species.

## Figures and Tables

**Figure 1 microorganisms-11-00971-f001:**
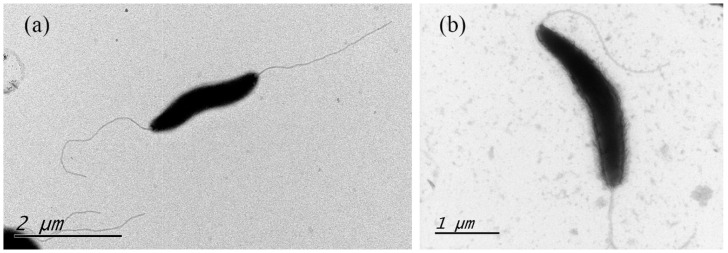
Transmission electron microscope image of the novel *Campylobacter* strains from 48 h culture. (**a**) *C. felis* strain XJK22-1^T^. (**b**) *C. ovis* strain SYS25-1^T^.

**Figure 2 microorganisms-11-00971-f002:**
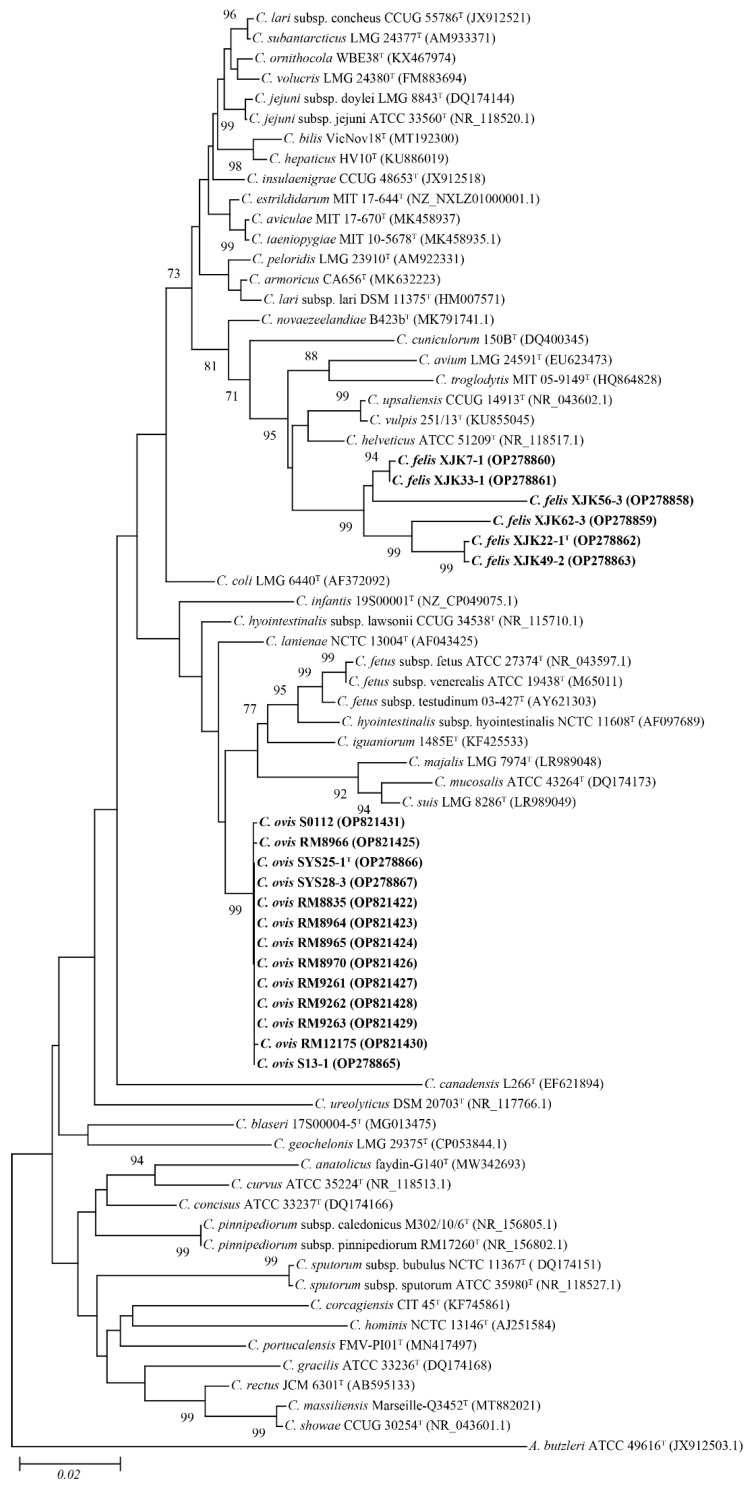
Neighbor-joining phylogenetic tree based on nearly complete 16S rRNA gene showing the relationships between our isolates and the type strains of the genus *Campylobacter*. Bootstrap values (>70%) based on 1000 replicates are shown at branch nodes, with *Arcobacter butzleri* ATCC 49616^T^ as an outgroup. Bar—0.02 changes per nucleotide position. Novel strains are highlighted in bold.

**Figure 3 microorganisms-11-00971-f003:**
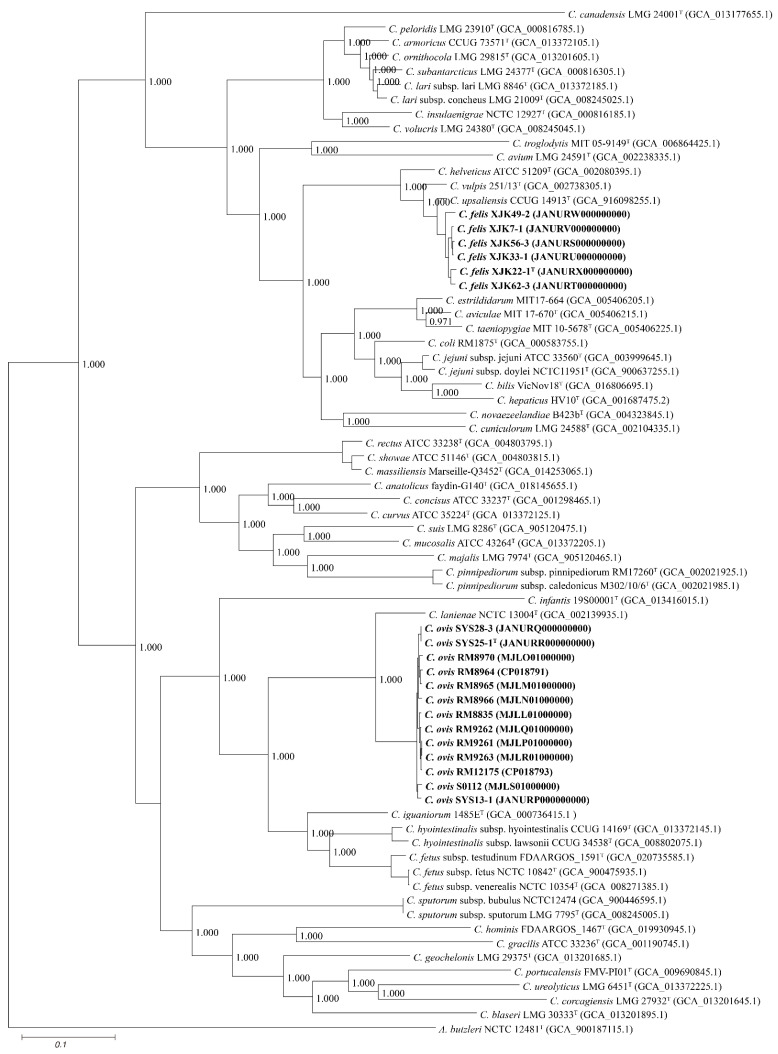
Neighbor-joining phylogenomic tree based on 332 core genes of the genus *Campylobacter*. The outgroup is *Arcobacter butzleri* ATCC 49616^T^. Novel strains are highlighted in bold.

**Figure 4 microorganisms-11-00971-f004:**
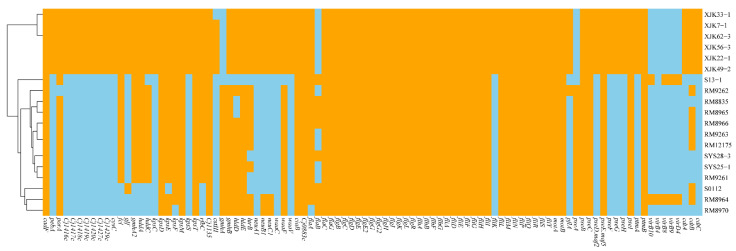
Heatmap of the distribution of virulence genes. Orange indicates the presence of the virulence genes, and sky blue indicates the absence of the virulence genes.

**Table 1 microorganisms-11-00971-t001:** Primers and probes used in this study.

Target	Sequence(5′–3′) ^a^	Target Gene/Region	Reference
*Campylobacter* spp.	27F: AGAGTTTGATCCTGGCTCAG	16S rRNA gene	[8]
	1492R: CGGTTACCTTGTTACGACTT		
*C. felis* sp. nov.	F: GCGCCATCTTGGACGAGTAT	Putative hydrolase YxeP gene	This study
	R: GGGCAGGGCGTCCATATC		
	P: FAM-CGCGAAGGAGGACGCAGGGA-BHQ1		
*C. ovis* sp. nov.	F: TGAAGCTGGAGAAAGTGGCC	Hypothetical protein gene	This study
	R: TCCTATTATGGCGCCAGCTG		
	P: FAM-CAACCCTAAGTAGCGGAAGCGGTGG-BHQ1		

^a^ FAM—6-carboxyfluorescein.

**Table 2 microorganisms-11-00971-t002:** Background of strains *Campylobacter felis* sp. nov. and *Campylobacter ovis* sp. nov.

Isolate	Species	Isolation Date	Isolation Country: City	Host	Source	16S rRNA Gene Accesion Number	Accession Number
XJK22-1	*C. felis*	2019	China: BeiJing	cat	feces	OP278862	JANURX000000000
XJK33-1	*C. felis*	2019	China: BeiJing	cat	feces	OP278861	JANURU000000000
XJK49-2	*C. felis*	2019	China: BeiJing	cat	feces	OP278863	JANURW000000000
XJK56-3	*C. felis*	2019	China: BeiJing	cat	feces	OP278858	JANURS000000000
XJK62-3	*C. felis*	2019	China: BeiJing	cat	feces	OP278859	JANURT000000000
XJK7-1	*C. felis*	2019	China: BeiJing	cat	feces	OP278860	JANURV000000000
S13-1	*C. ovis*	2020	China: BeiJing	Sheep	feces	OP278865	JANURP000000000
SYS25-1	*C. ovis*	2019	China: BeiJing	Sheep	feces	OP278866	JANURR000000000
SYS28-3	*C. ovis*	2019	China: BeiJing	Sheep	feces	OP278867	JANURQ000000000
RM8835	*C. ovis*	2009	USA: California	Alpaca	feces	OP821422	MJLL01000000
RM8965	*C. ovis*	2009	USA: California	Goat	feces	OP821424	MJLM01000000
RM8966	*C. ovis*	2009	USA: California	Goat	feces	OP821425	MJLN01000000
RM9262	*C. ovis*	2009	USA: California	Alpaca	feces	OP821428	MJLQ01000000
S0112	*C. ovis*	2013	UK: Scotland	Sheep	feces	OP821431	MJLS01000000
RM8970	*C. ovis*	2009	USA: California	Goat	feces	OP821426	MJLO01000000
RM9263	*C. ovis*	2009	USA: California	Alpaca	feces	OP821429	MJLR01000000
RM9261	*C. ovis*	2009	USA: California	Alpaca	feces	OP821427	MJLP01000000
RM12175	*C. ovis*	2010	USA: California	Alpaca	feces	OP821430	CP018793
RM8964	*C. ovis*	2009	USA: California	Goat	feces	OP821423	CP018791

**Table 3 microorganisms-11-00971-t003:** Phenotypic characteristics of *Campylobacter felis* sp. nov. strains, *Campylobacter ovis* sp. nov. strains, and the type strains of related species.

Species	Isolate	Catalase	Oxidase	URE	NIT	EST	HIP	GGT	TTC	PyrA	ArgA	AspA	PAL	H_2_S
*C. felis*	XJK22-1	−	+	−	+	+	+	−	−	−	+	+	+	−
*C. felis*	XJK33-1	−	+	−	+	+	+	−	−	−	+	+	+	−
*C. felis*	XJK49-2	−	+	−	+	+	+	−	−	−	+	+	+	−
*C. felis*	XJK56-3	−	+	−	+	+	+	−	−	−	+	+	+	−
*C. felis*	XJK62-3	−	+	−	+	+	+	−	−	−	+	+	+	−
*C. felis*	XJK7-1	−	+	−	+	+	+	−	−	−	+	+	+	−
*C. ovis*	S13-1	+	+	−	+	−	−	+	−	−	+	−	+	−
*C. ovis*	SYS25-1	+	+	−	+	−	−	+	−	−	+	−	+	−
*C. ovis*	SYS28-3	+	+	−	−	−	−	+	−	−	+	−	+	−
*C. upsaliensis*	CCUG 14913	−	+	−	+	+	−	−	−	−	+	+	+	−
*C. coli*	ATCC 33559	+	+	−	−	+	−	−	−	−	+	−	+	−
*C. concisus*	ATCC 33237			−	−	−	−	−	−	−	−	+	−	+
*C. gracilis*	ATCC 33236			−	−	+	−	+	−	−	+	+	−	+
*C. helveticus*	CPD4-1	−	+	−	+	+	+	+	−	−	−	−	−	−
*C. hyointestinalis*	ATCC 35217	+	+	−	−	−	−	+	−	−	−	−	+	−
*C. jejuni* subsp. doylei	ATCC 49349			−	−	+	+	+	+	+	−	−	+	−
*C. jejuni* subsp. jejuni	ATCC 33560	+	+	−	−	+	+	−	−	−	−	−	+	−
*C. lari*	ATCC 35221	+	+	−	+	−	−	+	−	−	+	−	−	−
*C. rectus*	ATCC 33238			−	−	+	−	−	−	−	+	+	+	−
*C. showae*	ATCC 51146			−	−	+	−	+	−	−	+	+	−	+
*C. jejuni* subsp. jejuni	NCTC 11168			−	+	+	+	−	−	+	−	−	+	−
*C. upsaliensis* ^a^	CCUG 14913	−	+	−	+	+	−	−	−	−	+	+	+	−
*C. lanienae* ^a^	CCUG 44467	+	+	−	+	+/−	−	−	+	−	+/−	−	+	−

^a^ These data are referenced from BacDave (https://bacdive.dsmz.de/) (accessed on 15 February 2023).

**Table 4 microorganisms-11-00971-t004:** Genomes characteristics of strains *Campylobacter felis* sp. nov. and *Campylobacter ovis* sp. nov.

Strain	Contigs	Bases	GC Content	CDS	rRNA	CRISPR	tRNA
XJK22-1	42	1,700,455	34.99%	1747	2	-	43
XJK33-1	53	1,666,017	35.05%	1707	2	-	43
XJK49-2	37	1,640,989	35.10%	1711	4	-	44
XJK56-3	39	1,664,700	35.03%	1693	2	-	42
XJK62-3	42	1,750,246	34.90%	1841	2	-	43
XJK7-1	51	1,668,760	35.04%	1710	3	-	44
S13-1	51	1,442,012	32.69%	1454	3	1	37
SYS25-1	10	1,580,362	32.43%	1567	1	1	38
SYS28-3	17	1,590,016	32.43%	1576	2	1	39
RM8835	49	1,694,176	32.21%	1722	2	2	39
RM8965	28	1,502,748	32.45%	1499	2	1	38
RM8966	62	1,608,325	32.33%	1601	2	3	39
RM9262	86	1,707,487	32.11%	1738	2	2	33
S0112	18	1,533,040	32.28%	1545	2	2	39
RM8970	44	1,496,869	32.43%	1493	2	1	32
RM9263	59	1,632,618	32.31%	1680	2	2	39
RM9261	66	1,634,929	32.31%	1673	2	2	39
RM12175	3	1,612,610	32.40%	1645	6	3	40
RM8964	2	1,754,294	32.08%	1771	6	1	41

**Table microorganisms-11-00971-t005a:** 

(A) *Campylobacter felis* sp. nov. Strains with Their Closely Related *Campylobacter* Species.									
	**XJK22-1**	**XJK33-1**	**XJK49-2**	**XJK56-3**	**XJK62-3**	**XJK7-1**	** *C. helveticus* **	** *C. upsaliensis* **	** *C. vulpis* **
XJK22-1		80.50%	73.80%	79.30%	82.20%	80.70%	31.90%	57.70%	39.60%
XJK33-1	97.70%		74.40%	94.00%	78.90%	93.60%	31.60%	58.10%	39.00%
XJK49-2	97.01%	97.10%		74.60%	74.30%	74.40%	31.30%	57.30%	38.90%
XJK56-3	97.60%	99.26%	96.98%		78.80%	91.30%	31.50%	58.30%	38.90%
XJK62-3	97.94%	97.60%	96.92%	97.60%		79.30%	32.90%	57.50%	38.70%
XJK7-1	97.78%	99.18%	97.01%	98.98%	97.62%		31.70%	58.10%	39.00%
*C. helveticus*	86.55%	86.60%	86.45%	86.59%	87.21%	86.61%		29.90%	28.50%
*C. upsaliensis*	94.45%	94.71%	94.54%	94.74%	94.58%	94.71%	85.53%		40.20%
*C. vulpis*	89.92%	89.74%	89.70%	89.74%	89.64%	89.77%	84.51%	89.90%	

**Table microorganisms-11-00971-t005b:** 

(B) *Campylobacter ovis* sp. nov. strains with their closely related *Campylobacter* species.																
	**S13-1**	**SYS25-1**	**SYS28-3**	**RM8835**	**RM8965**	**RM8966**	**RM9262**	**S0112**	**RM8970**	**RM9263**	**RM9261**	**RM12175**	**RM8964**	***C. hyointestinalis* subsp. lawsonii**	** *C. infantis* **	** *C. lanienae* **
S13-1		75.10%	75.00%	75.20%	74.20%	74.30%	75.60%	78.30%	74.70%	75.30%	75.30%	75.30%	74.30%	19.40%	21.50%	25.10%
SYS25-1	97.14%		99.60%	77.30%	81.50%	80.70%	78.10%	74.80%	80.60%	78.10%	78.10%	77.40%	79.60%	19.50%	18.60%	25.00%
SYS28-3	97.09%	99.91%		77.10%	81.30%	80.50%	78.00%	74.60%	80.50%	78.10%	78.00%	77.30%	79.50%	19.70%	19.30%	25.10%
RM8835	97.15%	97.34%	97.36%		76.90%	76.70%	88.20%	75.40%	77.50%	86.00%	86.00%	84.00%	75.10%	19.70%	20.90%	25.20%
RM8965	96.98%	97.84%	97.86%	97.41%		80.80%	77.60%	73.70%	83.30%	77.60%	77.60%	77.30%	83.00%	19.40%	21.60%	25.10%
RM8966	96.97%	97.80%	97.79%	97.31%	97.83%		77.90%	74.10%	80.80%	77.70%	77.60%	77.30%	81.20%	19.70%	19.90%	25.20%
RM9262	97.14%	97.43%	97.44%	98.61%	97.44%	97.34%		75.60%	78.20%	93.70%	93.70%	92.60%	75.80%	20.30%	22.10%	25.20%
S0112	97.51%	97.10%	97.12%	97.18%	96.99%	97.07%	97.24%		75.00%	75.00%	75.00%	75.30%	74.10%	19.80%	21.50%	25.00%
RM8970	97.03%	97.75%	97.78%	97.36%	98.02%	97.79%	97.47%	97.11%		78.30%	78.20%	78.00%	83.60%	19.30%	20.60%	25.10%
RM9263	97.15%	97.45%	97.44%	98.37%	97.48%	97.36%	99.25%	97.16%	97.54%		100.00%	95.30%	75.80%	20.00%	22.00%	25.10%
RM9261	97.10%	97.38%	97.40%	98.33%	97.46%	97.31%	99.22%	97.14%	97.44%	99.99%		95.30%	75.70%	20.00%	22.00%	25.10%
RM12175	97.15%	97.49%	97.53%	98.18%	97.47%	97.41%	99.16%	97.22%	97.51%	99.44%	99.43%		77.20%	20.10%	23.50%	25.20%
RM8964	97.06%	97.70%	97.72%	97.16%	98.02%	97.84%	97.31%	97.06%	98.04%	97.27%	97.27%	97.41%		19.60%	23.50%	25.20%
*C. hyointestinalis* subsp. lawsonii	72.25%	72.22%	72.34%	72.39%	72.21%	72.36%	72.44%	72.25%	72.13%	72.29%	72.31%	72.34%	72.17%		21.30%	22.80%
*C. infantis*	68.44%	68.23%	68.40%	68.26%	68.33%	68.24%	68.32%	68.38%	68.39%	68.26%	68.24%	68.34%	68.35%	68.09%		20.90%
*C. lanienae*	82.30%	82.36%	82.43%	82.44%	82.43%	82.34%	82.32%	82.27%	82.51%	82.32%	82.32%	82.34%	82.44%	74.01%	68.58%	

Note: The sequence used is the same as in phylogenomic analysis.

## Data Availability

Not applicable.

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
