# Peer review of "Isolation and Genomic Characteristics of Cat-Borne Campylobacter felis sp. nov. and Sheep-Borne Campylobacter ovis sp. nov"

_microorganisms, 2023, doi:10.3390/microorganisms11040971_

Round 1
Reviewer 1 Report
I had the pleasure of reading this paper, and found it very interesting indeed. Discovery of new strains of bacteria is always interesting, but the significance is often missed. I also think that is missing a bit here, and some further details in some places would greatly benefit this, and I have tried to detail this below
Unfortunately, there is no line numbers so I have tried to do my best to signpost the authors to where my points relate to
Line 1 of the abstract- I would be tempted to remove ‘independently’
1st paragraph of page 2- what is GBS?
Materials and methods- some details on the animals here would be good- particularly with regards clinical signs
First paragraph of page 3- specifics on the antibiotics – specific antibiotics and concentrations here would be good
Species specific PCR
Line 4 and 5, delete the before 30s
Line 10- which were stored in our lab …. May sound better?
Were any internal controls used for the qPCR, such as housekeeping genes to detect for inhibitors etc?
Genomic analysis- first sentence doesn’t make sense, so please reword
Top of page 4- why was the 16S rRNA gene not sequenced from a pure culture rather than clone it?
Were the sequences put through the NCBI Blast software?
Results and discussion- Hippurate- sometimes these have capital letters and others don’t- please be consistent
Page 5- penultimate line of first paragraph- this doesn’t make sense- please reword
Antibiotic resistance and pathogenicity- please talk about the concentration of the antibiotics
Real time PCR- is there any idea of the general prevalence of the bacteria?
Please check bacterial names are in italics in the legends of the figures.
Author Response
Line 1 of the abstract- I would be tempted to remove ‘independently’
Response: Thanks for your comments and suggestion. We've modified it according to your suggestion.
1st paragraph of page 2- what is GBS?
Response: Thanks for your careful review. We have modified all the abbreviations in the main text when they first appear.
Materials and methods- some details on the animals here would be good- particularly with regards clinical signs
Response: Thanks for your comments and suggestion. In this study, these two novel species were isolated from asymptomatic carriers in cats and sheep, and we have modified this MS according to your suggestion.
First paragraph of page 3- specifics on the antibiotics – specific antibiotics and concentrations here would be good
Response: Thanks for your comments and suggestion. We've supplemented specific antibiotics and concentrations according to your suggestion.
Species specific PCR
Line 4 and 5, delete the before 30s
Response: Thanks for your comments and careful review. We've changed it according to your suggestion.
Line 10- which were stored in our lab …. May sound better?
Response: Thanks for your comments and careful review. We've changed it according to your suggestion.
Were any internal controls used for the qPCR, such as housekeeping genes to detect for inhibitors etc?
Response: Thanks for your comments. The target fragments we selected for detection belong to the specific genes of these two strains. Meanwhile, we set up blank controls and multiple negative controls during the experiment. In addition, the identification of strains in this study was a qualitative experiment, so we did not add internal controls.
Genomic analysis- first sentence doesn’t make sense, so please reword
Response: Thanks for your comments and careful review. We've changed it according to your suggestion.
Top of page 4- why was the 16S rRNA gene not sequenced from a pure culture rather than clone it?
Response: Thanks for your comments. Our previous experience has shown that part of Campylobacter species may have intervening sequences (IVSs) at about 200bp site of 16S rRNA at different growth phases, which leads to the failure of nearly complete length sequencing due to heterozygous signals. So, we chose to clone it to the vector for sequencing.
Were the sequences put through the NCBI Blast software?
Response: Thanks for your comments. Yes, we selected the culture of suspected colonies for 16S rRNA amplification sequencing and NCBI Blast comparison, to conduct preliminary identification of suspected colonies. Thereafter, the clone sequenced 16S rRNA was compared with other Campylobacter species by EzBioCloud’s identification service to locate their taxonomic position. We've shown the highest identity according to your suggestion.
Results and discussion- Hippurate- sometimes these have capital letters and others don’t- please be consistent
Response: Thanks for your comments and careful review. We've changed it according to your suggestion.
Page 5- penultimate line of first paragraph- this doesn’t make sense- please reword
Response: Thanks for your comments and careful review. We've changed it according to your suggestion.
Antibiotic resistance and pathogenicity- please talk about the concentration of the antibiotics
Response: Thanks for your comments and careful review. We've modified it according to your suggestion.
Real time PCR- is there any idea of the general prevalence of the bacteria?
Response: Thanks for your comments. In this MS, we focus on these two species as novel species, and the prevalence survey of these two novel species will be carried out in future studies.
Please check bacterial names are in italics in the legends of the figures.
Response: Thanks for your comments and careful review. We've checked them according to your suggestion.
Reviewer 2 Report
This is an important paper highlighting the identification of two novel species of Campylobacter , Campylobacter felis and ovis isolated from cats and sheep respectively. The paper is well written and all the necessary experiments have been carried out. Authors have also performed phylogenetic analysis which is crucial in the identification of the novel species. A minor question is whether those two new species poses a threat in public health and whether they are pathogenic for humans or not. The fact that they have been identified in cats and sheeps rise such a question since both animals are in close contact with humans.
Author Response
This is an important paper highlighting the identification of two novel species of Campylobacter , Campylobacter felis and ovis isolated from cats and sheep respectively. The paper is well written and all the necessary experiments have been carried out. Authors have also performed phylogenetic analysis which is crucial in the identification of the novel species. A minor question is whether those two new species poses a threat in public health and whether they are pathogenic for humans or not. The fact that they have been identified in cats and sheeps rise such a question since both animals are in close contact with humans.
Response: Thanks for your comments and careful review. These two new species were discovered during the initial research project to understand whether Campylobacter derived from animals in close contact with humans posed a threat to public health. In this study, we focus on these two species as novel species, and experiments on animal experiments and the pathogenicity of these two novel species will be carried out in future studies.
Reviewer 3 Report
Comments to the authors for evaluating the following manuscript
Title: “Isolation and genomic characteristics of cat-borne Campylobacter felis sp. nov. and sheep-borne Campylobacter ovis sp. nov.”
· More criteria about the selected host (cat) must be added
· More information about physiological and culture characteristics of these novel strains are required
· Please added the overall practical implication of this study “ short conclusion” at the end of the abstract
· The information and references in the introduction section are very poor and there are new information about the campylobacter were published in the last five years. Additionally, it failed to explore the main problems and the aim and hypothesis of this study. Therefore, I suggest the rephrasing of the introduction sections by adding more and more information
· Although the novelty of this study, meanwhile the presentation of the results and discussion need further details and information. It should focus on explaining and evaluating what you found (the main results), showing how it relates to the new researches
· Please add at the end of the manuscript the limitations: what can’t the results and discussion tell us?
Author Response
More criteria about the selected host (cat) must be added
Response: Thanks for your comments and careful review. In this study, these two novel species were isolated from asymptomatic carriers in cats and sheep. The selected host includes health or diarrhea animals like dogs, cats, sheep, and pigs that are in close contact with humans. We have modified this MS according to your suggestion.
More information about physiological and culture characteristics of these novel strains are required
Response: Thanks for your careful review.
In this study, we preliminarily conducted fundamental physiological characteristic experiments on the culture, morphology, biochemistry, and antimicrobial susceptibility of these novel strains, and described the main characteristics, especially the culture conditions, in this MS. More experiments on the physiological characteristics of these two novel strains, including animal experiments and pathogenicity experiments, will be carried out in future studies.
Please added the overall practical implication of this study “ short conclusion” at the end of the abstract
Response: Thanks for your comments. We’ve modified this MS at the end of the abstract according to your suggestion.
The information and references in the introduction section are very poor and there are new information about the campylobacter were published in the last five years. Additionally, it failed to explore the main problems and the aim and hypothesis of this study. Therefore, I suggest the rephrasing of the introduction sections by adding more and more information
Response: Thanks for your comments and suggestion. We've supplemented more information according to your suggestion. The information is mainly about the prevalence and outbreak of Campylobacter in animals, especially pets.
Although the novelty of this study, meanwhile the presentation of the results and discussion need further details and information. It should focus on explaining and evaluating what you found (the main results), showing how it relates to the new researches
Response: Thanks for your comments and careful review. In this study, we focus on these two species as novel species and verify this hypothesis through genotypic, phenotypic, phylogenetic, and phylogenomic analyses. Meanwhile, according to your suggestions, we have modified the partial contents of the results and discussion to better explain the results.
Please add at the end of the manuscript the limitations: what can’t the results and discussion tell us?
Response: Thanks for your comments and suggestion. We've added the limitations at the end of the manuscript according to your suggestion. Identifying these strains as novel strains is a preliminary study. We haven’t yet determined their pathogenicity which will be confirmed by further experimental studies.
Round 2
Reviewer 1 Report
I wish to thank the authors for their efforts on the modifications to the manuscript, and wish them the best of luck for the future
Reviewer 3 Report
Thank you for providing the new version of this manuscript all my criticism were well managed